# The Interplay between Obstructive Sleep Apnea, Chronic Obstructive Pulmonary Disease, and Congestive Heart Failure: Time to Collectively Refer to Them as Triple Overlap Syndrome?

**DOI:** 10.3390/medicina59081374

**Published:** 2023-07-27

**Authors:** Prakash Banjade, Kamal Kandel, Asmita Itani, Sampada Adhikari, Yogendra Mani Basnet, Munish Sharma, Salim Surani

**Affiliations:** 1Internal Medicine, Manipal College of Medical Sciences, Pokhara 33700, Nepal; banjadeprakash486@gmail.com (P.B.); kandelkamal010@gmail.com (K.K.); 2Internal Medicine, Institute of Medicine, Kathmandu 44600, Nepal; asmitanioct26@gmail.com; 3Internal Medicine, Chitwan Medical College, Bharatpur 44200, Nepal; drsampadaadk@gmail.com; 4Internal Medicine, Patan Academy of Health Sciences-School of Medicine, Lalitpur 26500, Nepal; yogendramanibasnet@gmail.com; 5Division of Pulmonary, Critical Care and Sleep Medicine, Baylor Scott and White, Temple, TX 76508, USA; munish.sharma@bswhealth.org; 6Pulmonary, Critical Care & Pharmacy, Texas A&M University, College Station, TX 79016, USA

**Keywords:** obstructive sleep apnea hypopnea syndrome, chronic obstructive pulmonary disease, congestive heart failure, overlap syndrome

## Abstract

Background and objectives: Obstructive sleep apnea-hypopnea syndrome (OSAHS) and chronic obstructive pulmonary disease (COPD) are independently linked to an increase in cardiovascular disease (CVD). Only a few studies have been published linking the association between overlap syndrome and congestive heart failure (CHF). This review highlights the interplay between overlap syndrome (OSAHS-COPD) and CHF. Materials and methods: We thoroughly reviewed published literature from 2005 to 2022 in PubMed, Google Scholar, and Cochrane databases to explore the link between overlap syndrome and cardiovascular outcomes, specifically congestive heart failure. Results: Research indicates that individuals with overlap syndrome are more likely to develop congestive heart failure than those with COPD or OSA alone. Congestive heart failure is a common comorbidity of overlap syndrome, and it has a two-way connection with sleep-related breathing disorders, which tend to occur together more frequently than expected by chance. Conclusions: CHF seems to have a strong relationship with OS. Further research is required to understand the relationship between OS and CHF.

## 1. Introduction

In 1985, David C. Flenely first used the term “overlap syndrome” (OS) to characterize the combination of chronic obstructive pulmonary disease (COPD) and obstructive sleep apnea (OSA). The overall prevalence of COPD and OSA alone is high, and studies have revealed a higher prevalence of COPD coexisting with OSA [1,2,3]. Patients with OS often experience a significant drop in their blood oxygen levels during sleep. This nocturnal oxygen desaturation is closely associated with a higher incidence of pulmonary hypertension and more extensive changes in the structure and function of the right side of the heart compared to patients who have either obstructive sleep apnea or chronic obstructive pulmonary disease alone. Other cardiovascular complications of OS include systemic hypertension, coronary artery disease, arrhythmia, and CHF [4,5]. OS carries a greater clinical significance because it is associated with significant co-morbid conditions [2]. It is prudent that clinicians maintain a vigilant attitude towards the possibility of OS, as timely recognition and appropriate management are imperative to prevent potential complications. One of the commonly encountered complications or comorbidities in OS is CHF. Hence, discussing the interplay between these three-disease entities (OSAHS-COPD-CHF) is important as a potential “triple overlap syndrome.” Our goal is to encourage research and collaboration among various disciplines to address the challenges presented by interconnected chronic disorders. We propose the term “Triple Overlap Syndrome” and stress the need for a collective approach.

## 2. Methods and Materials

### 2.1. Study Inclusion Criteria and Search Strategies

We performed a literature search of PubMed, Google Scholar, and Cochrane databases to identify the literature reporting association of overlap syndrome with cardiovascular outcomes, particularly CHF (Figure 1). Literature published between 2005 and 2022 discussing congestive heart failure in patients with overlap syndrome were included in the review. Management of references and removal of duplicates were performed using Mendeley Reference Manager. 

Keywords used for the literature search were ‘Overlap Syndrome (OR (Chronic Obstructive Pulmonary Disease and Obstructive Sleep Apnea)) AND (Heart Failure OR Congestive Heart Failure OR CHF).’

### 2.2. Study Selection

We carefully reviewed several studies and found 206 studies relevant to our topic. After removing duplicates, we screened 188 studies to find the data we needed for our review. Of those, we excluded some studies that did not meet our criteria and ended up with 8 full-text articles to include in our study. We excluded studies that lacked relevant information, had inaccessible full-text articles, or contained similar data already obtained from other studies.

## 3. Results

A literature search (Table 1) suggests that patients with OS have significantly higher morbidity and mortality than COPD or OSAHS alone. CHF is strongly associated with OS and is the primary cause of mortality. Tang et al. [6] performed a cross-sectional study of 518 patients, including OS, COPD only, and OSAHS cases. The study only included patients aged between 18 and 80 years old. Patients with a life expectancy of less than a year, a past upper airway surgery for OSA, and severe COPD requiring respiratory ventilators were not included in the study. OS patients mostly had mild (39.13%) or moderate (34.78%) obstructive sleep apnea. Those with only OSA mostly had severe OSA (42.68%). During sleep testing, overlap syndrome patients had a higher apnea/hypopnea index and lower mean SpO_2_ compared to OSA-only patients. The severity of COPD was similar in the OS group and COPD-only group, however FEV1/FVC ratio was lower in the OS group. This study showed that in comparison to patients with OSA alone or COPD alone, OS patients had higher prevalence rates of heart failure (10.8 vs. 0.5 and 1.4%, respectively) and pulmonary arterial hypertension (31.1 vs. 4.5 and 17.1%, respectively (all *p* < 0.01). Adle et al. [7] performed a cross-sectional study using prospective data from the French National Sleep Apnea Registry. They included patients who completed sleep study and have valid spirometry measurements. The median FEV1 values (% predicted) were 97 and 75 in OSA and OS, respectively. Among patients with OS, severity of COPD was classified as GOLD 1 (41.1%), GOLD 2 (46.5%), and GOLD 3 and 4 combined (12.3%). The result shows that patients with OSA and OS have a significant burden of concomitant metabolic and cardiovascular conditions. The prevalence of heart failure was higher in the OS group than the OSA-only group (4.4% versus 2.2%; *p* = 0.01). The strength of this study is the large sample size, objective assessment, and subgroup analysis of COPD and OSA patients, however it is possible that there is a selection bias due to the fact that only a large subset of patients underwent pulmonary function tests, rather than the entire population referred for suspicion of OSA.

The retrospective cohort study by Tang et al. [8] in 6554 patients (OS 192) revealed that individuals with OS had worsening baseline features and a higher prevalence of cardiovascular diseases, such as heart failure, compared to individuals with COPD or OSAS only (adjusted hazard ratio (aHR): 3.067 (1.521–6.185); *p* = 0.002) and PH (aHR: 2.006 (1.005–4.004); *p* = 0.048). Among 43 heart failure patients, 28 had OS, 14 had COPD only, and 1 had OSAS. There was no statistically significant difference in severity of COPD and OSA when comparing these groups to the OS group; however, the OS group were more likely to require CPAP, suggesting more chance of developing comorbidities in this group. The incidence of heart failure was significantly higher in the OS group (35.4%) than the COPD group (8.9%) and OSA group (0.6%). This study provides insights regarding CHF in OS; however, this is a retrospective, single center study. Some patients were diagnosed as having OSA during treatment of coronary heart disease, which may have overestimated the risk of CHF in OSA patients. 

The prospective cohort study of 10,149 patients by Kendzerska et al. [9] reported CHF in 31.5% of OS patients, 17.5% of COPD-only patients, and 6.7% of OSA-only patients. The study suggests that people with COPD and OSAS may experience more severe hypoxemia, cardiac dysrhythmias, pulmonary hypertension, and right heart failure. Of 10,149 participants, 12% had COPD, 25% had nocturnal hypoxemia (defined as at least 10 min of sleep with SaO2 less than 90%), and 30% had AHI > 30. Individuals with COPD and nocturnal hypoxemia had higher risk of cardiovascular comorbidities (hazard ratio 1.91 (1.60–2.28)) with COPD and AHI greater than 30 showing the highest risk. In this research, it was discovered that there was a notable synergistic impact of COPD and OSA on women, but not on men. This implies that women may need more proactive measures to address nocturnal hypoxemia, a factor that can be modified, making them a high-risk group. This research ensured consistent scoring criteria throughout the study and relied on validated algorithms to identify diseases through administrative data. Additionally, a substantial sleep cohort based on clinical data was used, with a significant number of events and a relatively long follow-up period.

The systematic review by Czerwaty et al. [10] suggests that compared to COPD alone or OSA alone, the OS diagnosis considerably increased the likelihood of developing hypertension. The OS patient had a substantially higher occurrence of coronary heart disease (CHD), including myocardial infarction. Studies have suggested that coronary artery disease is the leading cause of CHF in OS. 

In a prospective, observational study conducted by Bhalla et al. [11], 77 consecutive patients with reduced LVEF and CHF were observed over 12 months. The study aimed to point out the relation between CHF and sleep disordered breathing. Of these 77 patients, 38 had an AHI of less than five, while 39 had an AHI of five or higher. The study found that many patients with CHF and reduced LVEF had an AHI value greater than five. Clinical findings of OSA were seen in 37 patients out of 39 with an AHI of 5 or more. Among these 37 patients, 24 were male (64.9%) and 13 were female (35.1%). The most common cause of CHF in patients with OSA was coronary artery disease (64.9%), followed by dilated cardiomyopathy (32.4%). 

On the other hand, patients with NYHA class 2 had a higher prevalence of OSA (51.4%) compared to NYHA class 3 (37.8%) and NYHA class 4 (10.8%). It is recommended that sleep apnea screening be implemented in the evaluation and follow-up of heart failure patients. However, it should be noted that the study’s assessment of sleep-disordered breathing (SDB) was limited to using a specific device and not complete lab polysomnography, so the prevalence and severity of SDB in the cohort may have been underestimated. Despite some limitations, this study gives important insights into the bidirectional relationship between CHF and OSA.

Sharma et al. [5] performed an observational study on 18 patients (7 OS, 11 COPD only) and found that compared to the COPD-only group, the overlap group’s RV mass index (RVMI) was greater (19 6 g/m^2^ vs. 11 6 g/m^2^, *p* = 0.02). Patients with COPD who are over the age of 18 and are at least in GOLD stage 2, and have a smoking history of at least 10 pack-years were considered for inclusion. They underwent overnight laboratory-based polysomnography to determine if they had OSA. Those with an apnea-hypopnea index (AHI) greater than 10 per hour were identified as having overlap syndrome. Adult COPD patients (GOLD stage 2 or higher) with at least 10 pack-years of smoking history were included. Overnight laboratory-based polysomnography was performed to test for OSA. Subjects with an apnea-hypopnea index (AHI) >10/h were classified as having overlap syndrome. When comparing the overlap syndrome group to the COPD-only group, the RV remodeling index (RVRI) was greater in the overlap syndrome group (0.27 0.06 vs. 0.18 0.08, *p* = 0.02). The severity of oxygen desaturation was correlated with the level of RV remodeling in overlap syndrome participants (R2 = 0.65, *p* = 0.03).

The study by Chen et al. [12] including 126 patients (95 OS, 31 COPD-only) also highlights the association between overlap syndrome and heart failure. The study suggests that compared to those with COPD alone, those with OS had worse left diastolic function and a higher risk of developing coronary heart disease and congestive heart failure. The severity of overlap between COPD and OSAHS was correlated with the degree of left cardiac diastolic dysfunction. Nocturnal hypoxia and increased LV mass index were observed, which play roles in the development of CHF. The left ventricular diastolic dysfunction was correlated with AHI and oxygen desaturation index. OS patients had more apnea-hypopnea and oxygen desaturation indicating strong risk of CHF.

## 4. Discussion

It is becoming increasingly clear through research that there is a strong connection between OSA, COPD, and CHF. While these conditions have their own distinct origins and clinical features, they often coexist due to shared risk factors such as age, obesity, smoking, and systemic inflammation. Moreover, each condition’s pathophysiological mechanisms can exacerbate the others. Thus, a vicious cycle of worsening symptoms and outcomes is created. This interplay involves many factors, including the impact of OSA on COPD exacerbations and how COPD can affect cardiac function in CHF patients. Additionally, both OSA and COPD can contribute to systemic inflammation, oxidative stress, endothelial dysfunction, and neurohormonal abnormalities, which substantially increase the risk of CHF. Furthermore, CHF itself is a risk factor for OSA.

The pathogenesis of cardiovascular outcomes, including CHF in OS, is multifactorial and characterized by a cascade of events, as discussed below. Hypoxia triggers a chain of oxidative stress, systemic inflammation, and vascular dysfunction, which is further compounded by sympathetic overactivity. The influence of modifiable risk factors such as obesity and smoking further exacerbates this process. 

### 4.1. Hypoxia in Triple Overlap Syndrome (OSAHS-COPD-CHF)

Patients with COPD experience decreased oxygen levels and breathing difficulties during sleep, especially during the REM (rapid eye movement) phase, due to relaxed intercostal muscles and limited chest wall movement. In contrast, patients with OSA encounter episodes of breathing cessation and reduced breathing primarily caused by the collapse of the upper airway, reduced pressure within the chest, and the activation of the sympathetic nervous system. This leads to frequent awakenings during the night and excessive sleepiness during the day [13]. Individuals who suffer from OS may encounter heightened levels of hypoxemia and hypercapnia. This is caused by a decrease in sensitivity of the respiratory center to both hypoxemia and hypercapnia stimulation. This decrease is a result of chronic hypoxia and hypercapnia, which are brought on by COPD. Additionally, intermittent nocturnal hypoxia and sleep deprivation caused by OSA may further exacerbate these symptoms [6]. The nocturnal drops in oxygen levels with an increase in carbon dioxide levels are more significant in patients with OS than those with COPD or OSA alone [13]. It is crucial to note that individuals diagnosed with OS exhibit greater levels of daytime hypoxemia and hypercapnia compared to their COPD or OSA counterparts. Hence, when combined with nocturnal hypoxemia, daytime hypoxemia and hypercapnia can potentially amplify the risk of developing CHF [6].

### 4.2. Consequences of Hypoxia on the Cardiovascular System

When cells experience low oxygen levels (hypoxia) as in congestive heart failure, it triggers the activation of a protein called HIF-1α. This protein is involved in various cellular processes, such as controlling cell death, regulating blood vessel constriction, managing energy usage, and promoting the growth of new blood vessels (angiogenesis). In addition to HIF-1α, hypoxia also activates another protein called HIF-2α, which is responsible for triggering the production of pro-inflammatory cytokines (molecules that promote inflammation). The resulting release of IL-1, IL-6, TNF-alpha, and reactive oxygen species (ROS) creates oxidative stress within the vessel wall. This oxidative stress is the driving force behind the development of atherosclerosis, as it initiates endothelial dysfunction (damage to the inner lining of blood vessels), promotes the breakdown of fats in the blood (lipid peroxidation), and sustains the disease process through angiogenesis [14,15,16]. 

Research has demonstrated that individuals who have both obstructive sleep apnea (OSA) and chronic obstructive pulmonary disease (COPD) exhibit elevated sympathetic activity and decreased parasympathetic activity, as evidenced by measurements of heart rate modulation. Additionally, this could promote an increase in cardiac morbidities [17]. Furthermore, overlap patients have more arterial stiffness than patients with OSA alone (Figure 2) [18]. 

### 4.3. Possible Common Contributing Factors for OSAHS, COPD, and CHF

Obesity is considered the key risk factor for OSA. Patients with COPD and OSA are more susceptible to nocturnal desaturation because of the narrowing of the upper airways caused by neck obesity. Truncal obesity may weaken the respiratory muscles and diminish the compliance of the chest wall, which could lead to ventilation-perfusion mismatches and ventilatory dysfunctions. Such disturbance in ventilatory capacity results in the retention of carbon dioxide, which can be detrimental to the cardiovascular system [19]. Cigarette smoking is a common risk factor for both COPD and OSA. Smoking can also increase oxidative stress and inflammatory mediators, increasing the pathophysiologic process that underlies CHF [20].

### 4.4. OSA in Congestive Heart Failure

Congestive heart failure (CHF) and sleep-related breathing disorders have a two-way relationship. If sleep-disordered breathing is left untreated, it can have a negative impact on the function of the left ventricle, ultimately leading to CHF [21]. According to a prospective study of 108 heart failure patients by MacDonald et al., 61% had a sleep-related breathing disorder. Of this group, 30% had OSA, and 31% had central sleep apnea [22]. Studies have shown that the prevalence of OSA is much greater in patients with CHF, ranging from 30% to 50% [22,23]. The mechanism behind Cheyne-Stokes breathing pattern caused by central sleep apnea in CHF differs from that of obstructive sleep apnea. The main pathophysiology of OSA in CHF patients is a nocturnal rostral fluid shift. During the day, fluid can build up in the lower extremities when standing up. However, when sleeping supine, this fluid mobilizes to the upper body, including the neck. This can cause swelling in the soft tissues around the pharynx and directly contribute to airway collapse and obstructive sleep apnea [24,25]. Patients with heart failure who have sleep-disordered breathing (SDB) tend to have a worse prognosis, with a higher mortality rate, in comparison to those who do not have SDB [26].

## 5. Limitation of the Study

The existing studies suggest that patients with OS are at greater risk of developing CHF. However, only limited studies are available which explain the association between these two. We could analyze only few papers and data to conclude the result. 

### Future Research Directions

Through an extensive literature search, we found a potential strong association between OS and CHF. However, several gaps remain in our understanding. We propose following future research directions to advance knowledge in this field. 

To better understand potential triple overlap, it is crucial to conduct large-scale epidemiological studies determining its prevalence, incidence, and associated risk factors. Longitudinal studies can be useful in identifying the causal factors and patterns of disease progression between OSA, COPD, and CHF.Conducting molecular studies, animal models, and in vitro experiments can aid in comprehending the shared pathophysiological pathways, such as chronic inflammation, oxidative stress, endothelial dysfunction, and autonomic dysregulation. Identifying specific biomarkers and genetic predispositions can facilitate prompt detection and targeted treatments.Standardized diagnostic criteria and comprehensive assessment tools are necessary to identify and characterize a potential triple overlap syndrome accurately. Research must focus on developing validated screening questionnaires, diagnostic algorithms, and imaging techniques that consider this association’s unique challenges.Researching the most effective ways to manage this situation is imperative. Future studies must determine the safety and effectiveness of interventions that target each aspect. Exploring new treatments, such as upper airway stimulation and upcoming medicine options, is highly recommended to determine if they can treat this condition.Long-term predictive studies are required to determine the impact of this suggested triple overlap on patient outcomes, including morbidity, mortality, and health-related quality of life.Future research should explore the effectiveness of educational interventions and shared decision-making approaches in managing patients with OS and CHF.

## 6. Conclusions

Based on the review of prevailing literature on OS, there seems to be a significant association between CHF and OS. The mutual existence of all three of these conditions can be a significant cause of morbidity and mortality. Compared to COPD-only and OSAHS-only groups, OS patients possibly have worse diastolic dysfunction, more right ventricular mass index, more severe oxygen desaturation, and more frequent CHF. Due to the significant interplay between these entities, the triple overlap of OSAHS, COPD (OS), and CHF might have to be viewed as a single spectrum in the right clinical context. Due to paucity of data, further literature is needed to establish this association with more clarity. Identifying patients at risk of concomitant prevalence of all three of these conditions is important to initiate a timely management strategy and prevent complications.

## Figures and Tables

**Figure 1 medicina-59-01374-f001:**
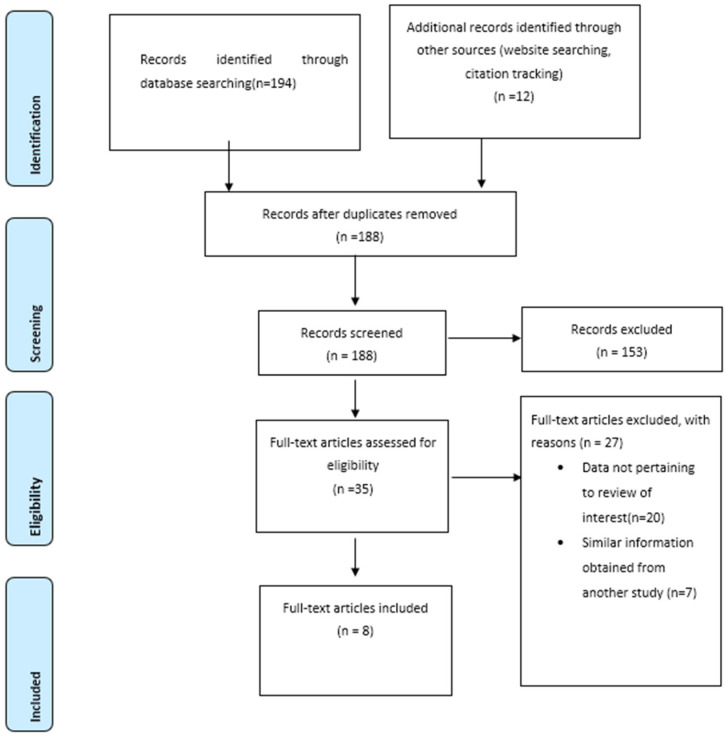
A PRISMA flow diagram depicting the literature search for OSAHS and CHF.

**Figure 2 medicina-59-01374-f002:**
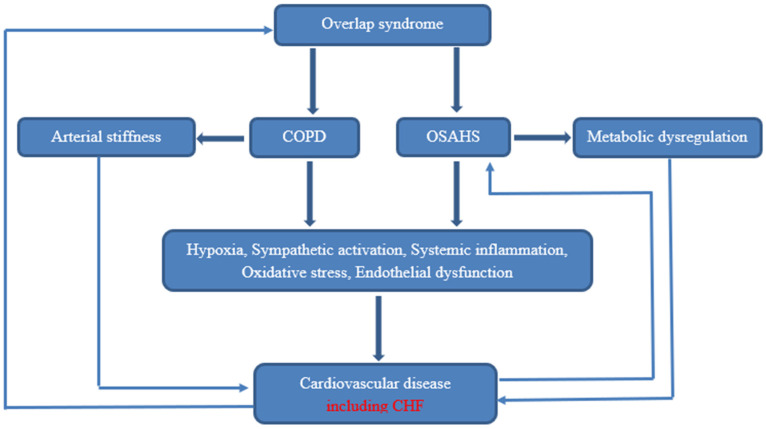
Pathophysiology of cardiovascular disease in OS.

**Table 1 medicina-59-01374-t001:** Relevant publications on OSAHS (overlapping sleep apnea-hypopnea syndrome) and CHF (congestive heart failure).

Author	Study Design	Study Country and Published Date	Sample Size	CHF Cases	Inferences
Tang et al. [6]	Cross-sectional	China, July 2021	Total = 518 OS = 74 COPD only = 222 OSA Only = 222	OS = 10.8% COPD only = 0.5% OSA only = 1.4%	Patients with OS have higher rates of heart failure (10.8%) and PAH (31.1%) compared to patients with only OSA (0.5% and 4.5%, respectively) or COPD (1.4% and 17.1%, respectively). These differences are statistically significant (*p* < 0.01).
Adle et al. [7]	A cross-sectional study utilizing prospective data from the French National Sleep Apnea Registry	France, July 2020	16,466	OS = 13% OSA = 87%	OS and OSA both possess significant burdens of concomitant metabolic and cardiovascular conditions. OS patients had a higher prevalence of heart failure (4.4% versus 2.2%; *p* < 0.01), stroke (4.3% versus 2.8%), coronary artery disease/myocardial infarction (13.4% versus 7.4%; *p* < 0.01), peripheral arteriopathy (6.0% versus 1.9%; *p* < 0.01), and hyperlipidemia (35.0% versus 29.7%; *p* < 0.01) when compared to OSA patients.
Tang et al. [8]	Retrospective cohort	China, July 2021	Total = 6554 OS = 192	Total = 43 OS = 28 COPD = 14 OSA = 1	Compared to patients with COPD or OSAS, people with OS had deteriorating baseline characteristics and a higher prevalence of cardiovascular illnesses, such as heart failure and pulmonary hypertension (aHR: 2.006 (1.005–4.004); *p* = 0.048) and heart failure (aHR: 3.067 (1.521–6.185); *p* = 0.002).
Kendzerska et al. [9]	Prospective Cohort	Canada, September 2018	10,149 OS = 5%	OS = 149 (31.5%) COPD only = 136 (17.5%) OSA only = 169 (6.7%)	The risk of cardiovascular disease and overall mortality was highest in people with COPD and nocturnal hypoxemia. People with OS may experience more severe hypoxemia, cardiac dysrhythmias, pulmonary hypertension, and right heart failure.
Czerwaty et al. [10]	Systematic review (38 studies)	Poland, December 2022	27,064	-	Compared to COPD alone or OSA alone, the OS diagnosis considerably increased the likelihood of developing hypertension. However, OSA was discovered to be a separate risk factor for hypertension. The OS patient had a considerably higher occurrence of coronary heart disease (CHD), including myocardial infarction.
Bhalla et al. [11]	Prospective cohort	India, November 2020	Total = 77 CHF cases	77	OSA was present in 50% of patients with CHF. Patients with LVEF 20–30% and NYHA class II were most likely to be affected.
Sharma et al. [5]	Observational	United States, February 2013	Total = 18 OS = 7 COPD only = 11		The patients with the overlap syndrome had a greater RV mass index (RVMI) than those with COPD only (196 g/m^2^ compared to 116 g/m^2^, *p* = 0.02). Additionally, the overlap syndrome group had a greater RV remodeling index (RVRI) than the COPD-only group (0.27 0.06 compared to 0.18 0.08, *p* = 0.02). The severity of oxygen desaturation was correlated with the level of RV remodeling in overlap syndrome participants (R2 = 0.65, *p* = 0.03).
Chen et al. [12]	Observational	China, February 2022	Total = 126 OS = 95 COPD only = 31		Compared to patients with COPD alone, patients with OS had worse left diastolic function and a higher risk of congestive heart failure. The severity of COPD overlapping sleep apnea-hypopnea syndrome was correlated with the degree of left cardiac diastolic dysfunction.

Abbreviations in the table—COPD: chronic obstructive pulmonary disease, OS: overlap syndrome, OSA: obstructive sleep apnea, PAH: pulmonary arterial hypertension.

## Data Availability

The data underlying this article are available within the article.

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
