# Peer review of "The Interplay between Obstructive Sleep Apnea, Chronic Obstructive Pulmonary Disease, and Congestive Heart Failure: Time to Collectively Refer to Them as Triple Overlap Syndrome?"

_medicina, 2023, doi:10.3390/medicina59081374_

Round 1

Reviewer 1 Report

A well written article. A few minor corrections needed. 

The results section seems bulky without proper paragraphing and is very difficult to read with all the different acronyms. Please rewrite. 

The discussion section can be elaborated a little further based on the 8 papers analysed, providing more insights on the results that the authors found. 

Authors need to acknowledge the limited papers and data found to conclude on the results and their findings. 

Author Response

1. The result section seems bulky without proper paragraphing and is very difficult to read with all the different acronyms. Please rewrite.

Response: We appreciate the reviewer’s comment. We have edited the result section with paragraphing and correcting unexpanded acronyms.

2. The discussion section can be elaborated a little further based on the 8 papers analyzed, providing more insights on the results that the authors found.

Response: We appreciate the reviewer’s comment. We have elaborated the discussion section further by adding ‘OSA in Congestive Heart Failure.’

3. Authors need to acknowledge the limited papers and data found to conclude on the results and their findings.

Response: We appreciate the reviewer’s comment. We have acknowledged the availability of limited studies in our limitation section of this review.

Reviewer 2 Report

Dear authors,

thank you for the opportunity to read and review the manuscript. The topic is very interesting.

General comments

The aim of the review is to outline how overlap syndrome is associated with a major risk of development of congestive heart failure. The aim is interesting. The review focuses on the cardiovascular complications in patients with overlap syndrome, e.g. arterial hypertension, ischemic heart disease and heart failure [van Zeller M, Basoglu OK, Verbraecken J, et al. Sleep and cardiometabolic comorbidities in the OSA-COPD overlap syndrome: Data from ESADA. ERJ Open Res 2023; in press (https://doi.org/10.1183/23120541.00676-2022)] but noy properly on congestive heart failure.

English language must be improved.

Specific comments

Title

The review does not properly focus on congestive heart failure but mostly on cardiovascular impairment in patients with overlap syndrome.

However, congestive heart failure can be the result of other cardiometabolic complications (as myocardial ischemia, hypertension, arrythmia, right heart failure, diabetes, obesity,  etc)

Abstract

Some abbreviations are missing.

English must be improved.

Methods and materials

Editing of the text is required.

Lines are missing throughout the paper.

What the exclusion of “information that was repeated from other studies” means? Different studies that used the same data?

Figure 1, the vertical boxes are not readable.

Results

English language must be improved.

Most of the studies cited report cardiovascular complications in overlap syndrome, just few of them focus on congestive heart failure.

Some abbreviations are missing.

Table 1

English must be improved.

Discussion

In the discussion section there is an interesting physiopathological description of the genesis of cardiovascular disease in overlap syndrome, it does not properly focus on congestive heart failure.

Conclusion

For the abovementioned reasons I think that conclusion is not properly consistent with the text.

Limitations section is missing.

Dear Authors, 

english language of the manuscript must be improved, mostly on abstract, table 1 and results section.

Author Response

1. The review does not properly focus on congestive heart failure but mostly on cardiovascular impairment in patients with overlap syndrome.

Response: We appreciate the reviewer’s insight and agree. There are limited number of overlap syndrome addressing the congestive heart failure and we have listed that as one of our limitations.

2. In the Abstract, some abbreviations are missing. English must be improved.

Response: We appreciate the reviewer’s comments. We have added the missing abbreviations and improved the English language. It has been once again reviewed by the English speaking person.

3. In methods and materials

a. Editing of the test is required

Response: We edited the text to give a better flow.

b. Lines are missing throughout the paper.

Response: We thank the reviewer for the comment. As per the comment, we have added lines throughout the paper.

c. What does the exclusion of “information that was repeated from other studies” mean? Different studies that used the same data?

Response: We appreciate the reviewer’s comment. To clarify the statement, we have changed the line to ‘We excluded studies that lacked relevant information, had

inaccessible full-text articles, or contained similar data already obtained from other studies.’

d. Figure 1, vertical boxes are not readable.

Response: We have corrected the image.

Reviewer 3 Report

This is an interesting review but misleading due to conclusions which are not supported by the included studies.

1) These are post-hoc comparisons of the prevalent cardiac failure in patients with OSA and COPD, and cardiac failure was not among the primary outcomes.

2) Patients with cardiac failure have usually central sleep apneas with Cheyne-Stokes pattern and these are different from obstructive sleep apneas and may coexist in patients with COPD.

3) The relationship between OSA and cardiac failure is bidirectional and there are other important mechanisms such fluid shift from legs to neck when lying during sleep and sleep-disordered breathing to include central apneas may fit better in the hypothesis generating review.

4) It is too early to call the co-existence of these 3 conditions as Triple Overlap Syndrome in the absence of prospective studies addressing this question.

Moderate editing is required.

Author Response

Please note: By error the replies from 4 to 7 are for reviewer 2....Then onwards its for Reviewer 3. We apologize for the error. System did not allow us to revert it

Continued for Reviewer 2...

Results

Reviewer’s comment:

English language must be improved.

Most of the studies cited report cardiovascular complications in overlap syndrome; just a few of them focus on congestive heart failure.

Some abbreviations are missing.

Response: We appreciate the reviewer’s comment. We have improved English, added studies describing congestive heart failure in overlap syndrome, and corrected the missing abbreviations.

5. Table 1: English must be approved

Response: We have improved the English language in the table.

6. Discussion: In the discussion section, there is an interesting physiopathological description of the genesis of cardiovascular disease in overlap syndrome; it does not properly focus on congestive heart failure.

Response: We appreciate the reviewers suggestion. We have added a section OSA in congestive heart failure. We thank the reviewer for that.

7. The conclusion is not properly consistent with the text. The limitation section is missing.

Response:

Comments on the Quality of the English Language

Reviewer’s comment: English language of the manuscript must be improved, mostly on the abstract, table 1, and results section.

Response: We appreciate the reviewer’s comment. We have improved the English language.

Reviewer 3 comments

1. These are post-hoc comparisons of the prevalent cardiac failure in patients with OSA and COPD, and cardiac failure was not the primary outcome.

Response: We appreciate the reviewer’s comment. We found only limited articles explaining congestive heart failure (CHF) as a primary outcome in patients with overlap syndrome. However, many studies have mentioned that CHF is significantly higher in patients with overlap syndrome than in COPD or obstructive sleep apnea

alone. We feel that the association is understudied and needs more research to come to a definite conclusion..

2. Patients with cardiac failure usually have central sleep apneas with a cheyne-strokes pattern and these are different from obstructive sleep apneas and may coexist in patients with COPD.

Response: We appreciate the reviewer’s comment. We have addressed this concern in the discussion section.

3. The relationship between OSA and cardiac failure is bidirectional and there are other important mechanisms such as fluid shift from legs to neck when lying during sleep and sleep-disorderd breathing to include central apneas may fit better in the hypothesis-generating review.

Response: We thank the reviewer for pointing out important issues. We have discussed the bidirectional relationship in the discussion section.

4. It is too early to call the co-existence of these 3 conditions as Triple Overlap Syndrome in the absence of prospective studies addressing this question.

Response: We agree that further studies are needed to conclude. However, the association between overlap syndrome and congestive heart failure is understudied.

Comment on the Quality of the English Language:

Moderate editing is required.

Response: We have improved the English language.

Round 2

Reviewer 2 Report

Dear authors,

thank you for the opportunity to read and review the paper.

The paper is well written, the revision was accurate and the quality of the manuscript has improved.

Reviewer 3 Report

The paper is improved. I have no further comments.

Fine.